# Liposome Formulation and In Vitro Testing in Non-Physiological Conditions Addressed to Ex Vivo Kidney Perfusion

**DOI:** 10.3390/ijms23147999

**Published:** 2022-07-20

**Authors:** Silvia Pisani, Enrica Chiesa, Ida Genta, Rossella Dorati, Marilena Gregorini, Maria Antonietta Grignano, Marina Ramus, Gabriele Ceccarelli, Stefania Croce, Chiara Valsecchi, Manuela Monti, Teresa Rampino, Bice Conti

**Affiliations:** 1Otorhinolaryngology Unit, Department of Surgical Sciences, Fondazione IRCCS Policlinico San Matteo, 27100 Pavia, Italy; silvia.pisani01@universitadipavia.it; 2Department of Drug Sciences, University of Pavia, Vle Taramelli 12, 27100 Pavia, Italy; enrica.chiesa@unipv.it (E.C.); ida.genta@unipv.it (I.G.); rossella.dorati@unipv.it (R.D.); 3Department of Internal Medicine and Therapeutics, University of Pavia, 27100 Pavia, Italy; mgregorini@hotmail.com; 4Department of Nephrology, Dialysis and Transplantation, Fondazione IRCCS Policlinico San Matteo, 27100 Pavia, Italy; ma.grignano@smatteo.pv.it (M.A.G.); m.r.m.ramus@smatteo.pv.it (M.R.); t.rampino@smatteo.pv.it (T.R.); 5Department of Public Health, Experimental and Forensic Medicine, University of Pavia, Via Forlanini 2, 27100 Pavia, Italy; gabriele.ceccarelli@unipv.it; 6Department of Clinical, Surgical, Diagnostic & Pediatric Sciences, University of Pavia, 27100 Pavia, Italy; 7Immunology & Transplantation Laboratory, Fondazione IRCCS Policlinico S. Matteo, 27100 Pavia, Italy; s.croce@smatteo.pv.it (S.C.); c.valsecchi@smatteo.pv.it (C.V.); 8Human Anatomy Unit, Department of Public Health, Experimental and Forensic Medicine, Histology and Embryology Unit University of Pavia, Biotechnology Laboratories Fondazione IRCCS Policlinico San Matteo, 27100 Pavia, Italy; manuela.monti@unipv.it

**Keywords:** drug delivery, liposomes, kidney, transplant, protein delivery

## Abstract

This work focuses on formulating liposomes to be used in isolated kidney dynamic machine perfusion in hypothermic conditions as drug delivery systems to improve preservation of transplantable organs. The need mainly arises from use of kidneys from marginal donors for transplantation that are more exposed to ischemic/reperfusion injury compared to those from standard donors. Two liposome preparation techniques, thin film hydration and microfluidic techniques, are explored for formulating liposomes loaded with two model proteins, myoglobin and bovine serum albumin. The protein-loaded liposomes are characterized for their size by DLS and morphology by TEM. Protein releases from the liposomes are tested in PERF-GEN perfusion fluid, 4 °C, and compared to the in vitro protein release in PBS, 37 °C. Fluorescent liposome uptake is analyzed by fluorescent microscope in vitro on epithelial tubular renal cell cultures and ex vivo on isolated pig kidney in hypothermic perfusion conditions. The results show that microfluidics are a superior technique for obtaining reproducible spherical liposomes with suitable size below 200 nm. Protein encapsulation efficiency is affected by its molecular weight and isoelectric point. Lowering incubation temperature slows down the proteins release; the perfusion fluid significantly affects the release of proteins sensitive to ionic media (such as BSA). Liposomes are taken up by epithelial tubular renal cells in two hours’ incubation time.

## 1. Introduction

Liposomes are classified as nanomedicines and they have been perhaps the most studied nano-drug delivery systems in the previous years, with the largest number of products on the market [1]. This is because they have been demonstrated to be versatile with a lot of advantages [2]. Among these, liposomes are able to alter the pharmacokinetics of loaded drugs, they can be functionalized on their surface in order to target specific cell membrane receptors and prevent unwanted interaction of the loaded drug with nontargeted tissues [3,4]. Moreover, liposomes can stabilize the loaded drug, enhance its circulation time and reduce its side effects. Their unicity resides in being vesicles, closer in composition to biologic components, such as cell membranes, and biocompatible. Recently, the use of liposomes in COVID-19 vaccine formulations corroborated to legitimize them as useful nano-drug delivery systems [5].

Schematizing, liposomes are vesicles consisting mainly of phospholipids and cholesterol. They are spherical in shape, composed of an aqueous internal compartment surrounded by a variable number of lipid bilayers. Liposomes’ formation is a spontaneous process that occurs when blends of phospholipids with cholesterol and/or other amphiphilic molecules are hydrated. Phospholipids form vesicles thanks to the hydrophobic effect of their acyl chains in the aqueous medium. The ties involved are hydrogen bonds, Van der Waals forces or other electrostatic interactions.

Liposomes’ properties are dependent on the characteristics of lipids, the degree of fatty acid saturation and cholesterol percentage. The main characteristics of liposomes are surface charge, size, shape, lamellarity, stability, encapsulation efficiency and molar ratio between components. A liposome diameter equal to or greater than 300 nm leads to a complement system activation, causing toxicity. The clearance of liposomes depends on liposome size. It was found that liposomes greater than 300 nm and less than 70 nm have shorter circulation time than the liposomes of size 150–200 nm [6,7]. The general trend is the smaller the particle, the longer the circulation time; 100 nm liposomes have a circulation time of 8–10 h, liposomes greater than 200 nm have a circulation time of 4–6 h and liposomes bigger than 400 nm show a circulation time of 1–4 h [8].

In these terms, liposomes have been proposed as a drug delivery system also for targeting kidney in vivo, with encouraging results. Lots of examples include liposomes’ use in kidney injury and refer to preclinical studies. Amphotericin B, an antifungal drug, vancomycin, an antibiotic, and some platin-based anticancer drugs have been encapsulated in liposomes and their nephrotoxicity has been reduced [9]. Liposomes loaded with anti-inflammatory steroidal drugs, such as prednisolone, were studied to prevent acute cellular rejection (ACR) in the early phase after kidney transplantation. The mechanism by which prednisolone causes its immunosuppressive and anti-inflammatory effects is mediated by genomic and nongenomic pathways. Prednisolone encapsulation in so-called “long circulating” PEGylated liposomes represents a successful strategy to prevent nontarget tissue side effects. These liposomes have dimensions of 100 nm and the phospholipid bilayer is coated with PEG, which allows them to circulate in blood for days upon intravenous injection. They prevent drug from diffusing over the endothelial lining of blood vessels and subsequently spreading over the body. These PEGylated liposomes are small enough to accumulate at inflamed sites where vascular permeability is enhanced [10,11]. Mannose-PEG-DSPE liposomes are another example. These liposomes were developed as long-term vesicles to treat diabetic nephropathy in a rat model. This new drug carrier has a high drug-loading ratio and can effectively target diabetic nephropathy. The fat-soluble 1,2-Distearoyl-sn-glycero-3-phosphorylethanolamine (DSPE) molecules embedded in the liposome bilayer, while water-soluble mannose exposed on the surface of liposomes improves the drug water solubility and can be identified by the high expression of carriers GLUT in the kidney to achieve targeted drug delivery. There was a GLUT saccharide recognition and binding site, so that the liposomes modified by mannose-PEG-DSPE could enter the renal cells through GLUT-mediated target. Sugar is bound with DSPE-PEG600 chains to form ligand compound. PEG has been demonstrated to prolong the circulation time by avoiding reticuloendothelial system clearance and facilitate the passive accumulation in tumor tissue [12]. Recently, the use of “empty” liposomes (no drug loaded) in dialysate during renal dialysis has been investigated, as scavengers both for exogenous intoxicants (intoxicants) and endogenous toxic molecules (endogenous toxins), which accumulate due to decreased excretion in cases of organ failure. Studies showed that intravascular “empty” liposomes, in vivo, act as detoxification vehicles [13].

Moreover, interesting examples of liposomes in clinical trials for different renal pathologies are reported in the clinical trials web site [14], as shown in Table 1.

Starting from this background that proves the effectiveness and versatility of liposomes as drug delivery systems for targeting kidney, this work focuses on formulating liposomes to be used in isolated kidney dynamic machine perfusion (MP) in hypothermic (HMP) conditions. The final goal is to supplement growth factors (i.e., hepatocyte growth factor) and/or ATP in order to improve preservation of harvested kidney. The need mainly arises from use of marginal donor kidney to improve the pool of disposable organ for the transplantation [15]. Marginal donors or expanded criteria donors (ECD) include older subjects with comorbidities, such as hypertension, mild renal impairment, and donors after circulatory death (DCD). It is known that kidneys from ECD and DCD are more exposed to ischemic/reperfusion injury (IRI) compared to those from standard donors. Scarce literature is reported on transplantation after ex vivo drug administration targeting the biological pathway associated to kidney failure (i.e., oxidative stress, complement system and fibrosis). Recently, extracellular vesicles (EV) from mesenchymal stem cells have been proposed as carriers of a pool of biologic factors that can improve renal survival during ex vivo perfusion [13,16,17]. Since liposomes have been demonstrated to be effective drug carriers for kidney targeting, in this preliminary work, we want to explore the ability of formulating liposomes that would be stable in non-physiologic conditions (namely hypothermic conditions and perfusion fluid) for safe and effective delivery to endothelial tubular cells. The hypothesis is that they might be a promising therapeutic strategy to improve the quality of various donor organs and expand organ availability. Moreover, liposomes might be an alternative to EV.

## 2. Results

### 2.1. Liposome Physical–Chemical Characterization

Table 2 reports the results in terms of average particle size, PDI, zeta potential and encapsulation efficiency of the liposome batches prepared with the two manufacturing techniques with the optimized liposome composition. The results of preliminary formulation screening are reported in the Appendix A.

Liposomes obtained by the TFH method resulted in always being bigger than those obtained with the microfluidic technique. BSA loading into liposomes was challenging by both preparation methods: it caused excessive liposome enlargement that quadrupled or doubled up their size, with always extremely low encapsulation efficiency (see Table 2, batches # 1 BSA and 2 BSA). It can be speculated that BSA loading promotes liposome swelling. Moreover, a higher polydispersity index (PDI) for BSA-loaded liposomes was obtained compared to the placebo liposomes obtained with both techniques. As far as liposomes loaded with Myo are concerned, results always show higher EE% with respect to liposomes loaded with BSA, and the highest EE% was obtained for liposomes prepared by TFH (Table 2, batch #1 Myo). The last result might be correlated with greater liposome size of liposomes obtained by TFH, with respect to Myo-loaded liposomes produced by microfluidics. As expected, all liposomes analyzed for their Z-potential confirmed their almost neutral charge independently of the protein loaded.

Figure 1 reports, as an example, TEM images of liposomes from batches #1 and 2. TEM analysis of all liposome batches confirmed the results of DLS analysis, showing that liposomes produced with TFH have a larger diameter than those produced with the microfluidic technique (see Table 2 and Table 3). The TEM images were processed with the ImageJ software and the data obtained as size and circularity are shown in Table 3 for the placebo liposomes. Liposome size obtained from TEM images by ImageJ analysis were further compared with those obtained with DLS, and thickness of the lipid bilayer was calculated from TEM/DLS diameter ratios (see Table 3).

Bilayer thickness measures showed no significant differences between liposomes obtained with both techniques. Liposomes have a spherical shape proved by values of circularity obtained through ImageJ processing. Values close to 1 identify a perfect spherical shape, while values near to 0 refer to a flatter shape. Liposomes produced with the microfluidic technique showed a more spherical shape compared to liposomes obtained through TFH.

The results of in vitro release tests reported in Figure 2 compare BSA and Myo release from liposomes produced by the two preparation methods tested. The in vitro release test was conducted in PBS at 37 °C, which are the conventional in vitro conditions simulating the physiologic environment. The goal of this preliminary test was to compare liposomes obtained by the different preparation methods. Time set was 4 h, since this is an average conditioning time of an ECD/DCD kidney in the perfusion machine. In these conditions, the graph shows how batch #1BSA produced with TFH shows a slower release rate with respect to those produced with Nanoassemblr, with an amount of 54.1 ± 2.7% of BSA released in 4 h, while batch #2BSA released about 66.0 ± 3.3% of the encapsulated protein in the same time frame. BSA in vitro release study was performed for a further 48 h and the amount of BSA remained in the liposomes after the in vitro release study (48 h) was evaluated, recovering pellets and treating them with Triton X solution 2% *v*/*v*. Results confirmed that the amount of BSA that is missing in the release after 48 h remained embedded in the liposomes. As far as Myo release was concerned, Batch #1Myo reached a 100% release in 4 h, while batch #2Myo achieved complete release already after 3 h, as shown in Figure 2. The Myo release profiles confirm that: (i) Myo release rate from batch 2Myo is significantly faster that Myo release rate from Batch 1Myo; (ii) Myo is released completely from the liposomes. Moreover, (iii) BSA release is significantly slower and incomplete if compared to Myo release.

The results obtained from preliminary characterization highlighted that liposomes obtained through microfluidic techniques showed smaller dimensions, rounder shape and higher process yield compared to liposomes obtained with TFH. BSA- and Myo-loaded liposomes obtained with microfluidic techniques showed smaller dimension and PDI compared to the loaded liposomes obtained through TFH. EE% calculated for both liposome types did not show significant differences in BSA loading, while Myo loading was lower for liposomes obtained by microfluidic technique. In vitro release test performed at 37 °C showed that faster drug release was achieved by liposomes obtained with the microfluidic technique. Considering our purpose to obtain liposomes smaller than 200 nm, with a PDI lower than 0.3 [6] and able to guarantee greater release in 4 h, only liposomes obtained through the microfluidic technique were further studied in order to enhance EE% and evaluate protein release in non-physiologic condition mimicking kidney perfusion (4 °C, 4 h). Therefore, liposome formulation and process parameters were modified and the results are reported in Figure 3 in terms of BSA EE%.

The results reported in Figure 3 show that 20% w/w trehalose addition in the aqueous protein solution enhances BSA encapsulation efficiency. Moreover, batches #9BSA and #10BSA, obtained using BSA dissolved in purified water, showed lower dimensions compared to batches #5BSA and #6BSA prepared with the same condition but using BSA dissolved in PBS. Batch #10 exhibits the best properties combining encapsulation efficiency (4.0 ± 1.3%) and particle size (139.8 ± 5.4 nm). These results were confirmed in placebo liposomes produced in the same conditions as batch# 10BSA, whose average particle size was observed to be 122.4 ± 3.8 nm. Batch #10BSA was used for further in vitro release studies (Figure 4) in physiologic simulated conditions (in PBS and 37 °C) and harvested organ simulated conditions (in PERF-GEN and 4 °C).

The results obtained show that Myo (batch #2 Myo) at 4 °C is fully released within 4 h, both in PBS and in PERF-GEN. Further, its release profile significantly changes with temperature, independently of the release medium. As shown in the graphs of Figure 4, Myo release is 80% in 1 h at 37 °C and it is about 58% and 45% at 4 °C both in PBS and in PERF-GEN. The results demonstrate that lowering temperature Myo release slows down in the first hour of incubation, either in PBS or in PERF-GEN. BSA (batch # 10BSA) reached a release peak of about 60% in PBS 37 °C and 40% in PBS 4 °C in the first hour, and then remained constant in the remaining 4 h, not releasing the whole content that remained trapped in the liposome. In PERF-GEN BSA, release is 60% after 1 h up to 70% BSA released in 4 h; this means that the perfusion liquid is able to increase the amount of BSA released by the liposome at 4 °C. Comparing the release profiles of Myo and BSA, it can be stated that both release profiles are modified by temperature, while PERF_GEN perfusion fluid modifies the release profiles of BSA. It significantly increases the release amount and release rate of BSA that has higher Mw with respect to Myo and is sensitive to ions, while it does not affect Myo release that has smaller Mw and is neutral, not sensitive to ions in the release medium.

### 2.2. Fluorescent-Labeled Liposomes In Vitro Test on Porcine Epithelial Tubular Renal Cells

The aim of in vitro testing of fluorescent porcine epithelial tubular renal cells co-incubated with fluorescent liposomes was to evaluate their cell uptake. Figure 5 shows a stack image of liposomes after 4 h with tubular renal cells and, through orthogonal-view image processing by ImageJ, it is possible to detect that the liposomes are internalized by cells.

Z-stack images processed by ImageJ software allow an orthogonal view to be obtained and location of the liposome in the cell using the position of the axes XY and YZ. The image selected (Figure 5) confirms the presence of fluorescent liposomes inside the cells. In particular, the liposomes inside the cell wall can be seen both on the XY and YZ axes.

Figure 6 shows time frame images of the liposomes incubated until 4 h with tubular renal cells. Red and yellow round markers highlight liposomes’ position inside cells. Highlighted liposomes follow cells movements during incubation time; the images confirm liposomes internalization already after 1 h.

### 2.3. Fluorescent-Labeled Liposomes Ex Vivo Characterization in Perfused Kidney from a Swine DCD Model

The samples of renal tissues from a pig DCD model were analyzed with a fluorescence microscope, after being fixed in paraffin and washed. t_0_ tissue (not treated with fluorescent liposomes) does not show autofluorescence with the filter used for liposome detection (Figure 7a,b). Tissue samples after 4 h of perfusion with liposomes show fluorescence compatible with the presence of liposomes in tissue portions around the vessels.

## 3. Discussion

Liposomes prepared by TFH and microfluidic techniques were compared in this study, confirming that the microfluidic technique is an advantageous method to obtain spherical liposomes with uniform and reproducible size. Comparing the morphometric results of TEM and DLS analysis, it can be stated that liposome bilayer thickness is in keeping with what is reported in the literature, taking into consideration that molecular weight of building compounds is 10^2^ [18]. This result corroborates what is reported in the literature [19,20,21]. The liposomes were loaded with two different proteins, namely myoglobin and bovine serum albumin. These can be models of protein drugs. In particular, BSA has a molecular weight similar to hepatocyte growth factor that is reported to be a pleiotropic factor playing a fundamental role in tubular repair and regeneration after acute renal injury [22,23].

As far as encapsulation is concerned, higher values were obtained with Myo with respect to BSA. This result can be due to the difference in molecular weights between the two molecules, BSA being Mw 66 kDa and Myo Mw 17 kDa. Moreover, carrying out encapsulation process PBS (pH 7.4), myoglobin (I.P. 7.0) has a neutral charge, while BSA (I.P. 4.7) is negatively charged. For this reason, Myo incorporation inside the lipid bilayer is not induced by charge interaction but only by physical incorporation, while BSA encapsulation is ion-specific and it is affected by medium ionic strength [24]. The same happens in in vitro release test, where Myo is completely released in 3 or 4 h of incubation, while BSA is not, even after 48 h release test. For this reason, the formulation of BSA-loaded liposomes was modified by the addition of trehalose to BSA solution, with the positive result of increasing four times BSA encapsulation efficiency. According to the literature, sugars can interact with the headgroups of the phospholipid, leading to formation of H bonds, which causes a depression of phase transition temperature (Tm), thus favoring encapsulation process [25]. Moreover, it is known from the literature that sugars, such has trehalose, have a positive impact in liposome storage stability [26].

Protein release from liposomes in the perfusion fluid PERF-GEN and at 4 °C, to the best of our knowledge, has not been tested and reported in the literature yet. Myo, which has smaller dimensions (17 kDa) and is neutral, is released entirely by the liposomes in 4 h at 4 °C both in PBS and in Perf-GEN. As expected, the temperature slows down Myo release rate because, at this temperature, lipid bilayer is more tight and rigid, being below phospholipid Tm. Thus, it can be stated that release of encapsulated Myo is regulated mainly by a diffusion process through lipid bilayer. BSA, on the other hand, has more difficulty in being completely released from the liposomes due to its larger molecular weight (66 kDa) and ion sensitivity. Furthermore, as already mentioned, BSA tends to dimerize in PBS, with an increase in molecular weight (132 kDa) [27] that might hamper its diffusion through the lipid bilayer. Moreover, as already mentioned, 4 °C is far below the lipids’ Tm, and liposome structure is more rigid and less flexible to allow BSA to diffuse through the phospholipid bilayer. BSA release was observed to be slightly improved in PERF-GEN (70% protein released in 4 h) and it is possible to hypothesize that the components of perfusion fluid, such as sugars, salts and amino acids, interact with the protein, forcing its diffusion from the liposome, even at low temperatures.

Cell uptake has been shown to happen in vitro, while it was poor in the ex vivo experiment. These preliminary results could be justified by the neutral surface charge of the liposomes, which delayed their interaction with cells that possess a negative surface charge [28], but also by the reduced quantity of liposomes used for perfusion (3 × 10^11^ liposomes/mL). Perhaps the use of ionizable lipids in the formulation of liposomes and the optimization of the number of liposomes to be used to have adequate cell penetration could improve the results. Furthermore, an ischemic kidney has damaged vascular tissue and the distribution and release of liposomes is more difficult in the first hours of perfusion, when renal flows and resistances are still high. However, we tested a new strategy for drug delivery to the kidney during dynamic PD and demonstrated that liposomes can be considered good vehicles, even under hypothermic conditions. Although the liposomes did not penetrate the cells massively, they remained in the renal tubular interstitium and, therefore, they could potentially act on various target cells: tubular, endothelial and fibroblast. Furthermore, they have proved to be a promptly usable vehicle, with the advantage of cell-free therapy, and are, therefore, suitable for the purpose of a reconditioning treatment before transplantation.

## 4. Materials and Methods

### 4.1. Materials

Chloroform reagent (RPE), methanol reagent (RS), ethanol reagent for analysis and sodium bicarbonate, purity 99.8%, were from Carlo Erba, Milan, Italy. Cholesterol, Mw 386.65 g/mol, dimethyl sulfoxide (DMSO), title 99.94%, myoglobin from equine skeletal muscle Mw 17,000 Daltons, 95–100%, essentially salt-free, lyophilized powder, phosphate buffer (PBS) tablets, trypsin-EDTA solution 0.25% and bovine serum albumin (BSA) Mw 69,234 Daltons, were from Sigma Aldrich, Milan, Italy. DSPC (1,2-distearoyl-sn-glycero-3-phosphocholine) Mw 790.145 g/mol was from Avanti-polar lipids Inc., Alabaster, United States. Fetal bovine serum (FBS) was from Euro Clone, Milan, Italy. Fluorescein-DHPE(N-(Fluorescein-5-Thiocarbamoyl)-1,2-Dihexadecanoyl-sn-Glycero-3-Phosphoethanolamine, Triethylammonium Salt), Mw 1182.54 g/mol (excitation/emission maxima ~496/519 nm) and MTT (3-(4,5-Dimethylthiazol-2-yl)-2,5-Diphenyltetrazolium Bromide) were from Thermofisher scientific, Waltham, Massachusetts, United States. Growing medium Dulbecco’s modified Eagle’s medium (DMEM) was from Microgem Laboratory Research, Milan, Italy. Penicillin and streptomycin solution 100X was from Dominique Dutscher, Brumath, France. Trypan Blue Solution w/v in PBS 0.4% w/vin normal saline (8.1 g/L NaCl with 0.6 g/L K_2_HPO_4_) was from Corning, United States. Double distilled water was produced by Milli-Q water purification system. All other reagents and solvents were of analytical grade, unless specified. PERF-GEN™ solution (composition: calcium chloride (dehydrated) 0.068 g/L, sodium gluconate 17.45 g/L, adenine 0.68 g/L, potassium phosphate (monobasic) 3.4 g/L, hydroxyethyl starch (HES) 50.0 g/L, magnesium gluconate 1.13 g/L), HEPES (4-(2-hydroxyethyl)-1-piperazineethanesulfonic acid) 2.38 g/L, glucose, beta D (+) 1.80 g/L, glutathione 0.92 g/L, ribose D (−) 0.75 g/L, mannitol 5.4 g/L, sodium hydroxide and sterilized water for injection were from Sigma-Aldrich Milan, Italy.

### 4.2. Methods

#### 4.2.1. Liposome Preparation

Liposomes were prepared by thin-film hydration (TFH) method, followed by extrusion and by microfluidic technique with Nanoassemblr apparatus. They were formulated with 1,2 distearoyl-sn-glycero-3-phosphocholine (DSPC) and cholesterol (Chol) with a molar ratio of DSPC: Chol 50:50, since preliminary experiments showed this was the more stable composition (see Appendix A by both preparation methods tested. The liposome compositions showing the best properties in terms of stability and particle size were selected for further encapsulation of either bovine serum albumin (BSA) or myoglobin (Myo). These proteins were used as models of protein drugs, such as growth factors (i.e., hepatocyte growth factor), with different molecular weights.

Liposome preparation by TFH was carried out as follows. The stock solutions of DPSC (10 mg/mL) and cholesterol (10 mg/mL) were prepared by solubilizing the lipids with a mixture of chloroform:methanol 9:1 volume ratio. Different volumes were taken from each stock solution to achieve the exact lipid DSPC:Chol 50:50 molar ratio and placed inside the round-bottom flask. The organic solvent was evaporated under vacuum using Heidolph^®^ Hei Vap Rotavapor (Heidolph, Germany) to obtain a homogeneous dry lipid film around the bottom flask wall. Then, the dry lipid film was rehydrated with phosphate-buffered saline (PBS) solution (pH 7.4) at 45 ± 3 °C to obtain placebo liposomes. Temperature set up (45 ± 3 °C) was chosen because it is close to DSPC (55 °C) and cholesterol (37 °C) main transition temperatures (Tm), making lipid chains more mobile and flexible. A round-bottom flask filled with PBS solution (1.3 mL) was vortexed (Advanced Vortex Mixer Zx3 Velp scientifica^®^, Italy) for 3 min at 24,000 rpm to suspend the lipid film and, subsequently, it was sonicated (SONICA^®^ ultrasonic cleaner Soltec^®^, Milan, Italy) for 3 min at 36–37 kHz to reduce the liposome size. Subsequently, the round-bottom flask with liposomes suspension was left 30 min in a warm water bath at 45 ± 3 °C to improve hydration and promote liposome formation. Finally, liposome suspension was manually extruded under mild heating (35 ± 3 °C) to produce a homogenous population of unilamellar liposomes through three polyvinylidene difluoride (PVDF) filter membranes of 0.45 µm, 0.22 µm and 0.1 µm. A total of 10 extrusion cycles were performed for each filter. Each filter was connected to two 5 mL (Luer-lock) syringes. The same procedure described above was applied to obtain myoglobin- and BSA-loaded liposomes, but, during rehydration phase, Myo solution in PBS buffer (0.15 mg/mL) or BSA solution in PBS buffer (10 mg/mL) were added. Figure 8a schematizes the thin-film hydration methods followed for the preparation methods of BSA- and Myo-loaded liposomes.

NanoAssemblrTM platform provided by Precision NanoSystems Inc. (Vancouver, Canada), characterized by a staggered herringbone micromixer (SHM), was employed for the liposomes’ preparation by microfluidic technique [29]. Micromixer cartridge dimensions are 6.6 × 5.5 × 0.8 cm (w × d × h); it is made of polypropylene, viton and cyclic olefin copolymer. The cartridge’s mixing channel is 200 × 79 μm (w × h) and the herringbone structure is 31 μm high and 50 μm thick. There is an angle of 45 between the ridges and the long axis of the channel. The microfluidic device consists of a Y-junction, known as a staggered herringbone, followed by a staggered mixing region. The staggered herringbone structures induce rapid mixing by chaotic advection [30]. Channel 1 was loaded with PBS (pH 7.4) or BSA solution in PBS (10 mg/mL) or Myo solution in PBS buffer (0.15 mg/mL), while DSPC:Chol 50:50 ethanol solution was loaded in channel 2 (see Figure 8b). Both inlet streams are controlled by syringe pumps connected to a computer that controls the process [31]. Stock solutions of DSPC (4.3 mg/mL) and cholesterol (2 mg/mL) were prepared by solubilizing lipids in ethanol (3 mL). Liposome preparation was carried out using a total flow rate (TFR) of 8 mL/min and a flow rate ratio (FRR) of aqueous phase: lipids phase 3:1 for 1 mL total volume. The preparation process was performed at room temperature (25 ± 3 °C). Preliminary experiments moving TFR from 8 mL/min to 12 mL/min and FRR between 5:1 and 3:1 were carried out to set up the process parameters (see Appendix A.

In order to increase BSA EE%, BSA loading into liposomes prepared by microfluidic technique was also investigated, modifying the formulation and process parameters as follows:
Liposomes’ lipid molar ratio was changed from 50:50 to DSPC:Chol 70:30;Total flow rate (TFR) was increased from 8 mL/min to 12 mL/min;Different amount of trehalose (10% *w*/*w*, 20% *w*/*w* and 40% *w*/*w*) were added to BSA aqueous solution to increase ionic strength.BSA was solubilized in purified water (pH 7.4 adjusted by NaOH 0.1 M) to prevent its dimerization and maintain Mw 66 kDa.

The batches produced are reported in Table 4.

#### 4.2.2. Liposome Characterization

Liposomes obtained by TFH and microfluidic techniques were characterized for their dimension, polydispersity index (PDI) and Z-potential using dynamic light scattering (DLS). Morphological characterization was performed through transmission electron microscopy (TEM), encapsulation efficiency (EE%), and Myo and BSA in vitro release test. The preliminary physical–chemical characterization showed superior properties of liposomes manufactured by microfluidic technique. Therefore, the placebo liposomes manufactured by microfluidic technique were fluorescent-labeled and underwent further characterization in vitro on epithelial tubular renal cell cultures and ex vivo on perfused harvested pig kidney.

DLS analysis was performed with Nicomp 380ZLS (Particle Sizing Systems, Lakeview Blvd. Fremont, CA, USA) on 300 µL of freshly prepared liposome suspensions in PBS buffer. The main parameters set up were: channel 10, intensity 100 kHz, temperature 23 °C, viscosity 0.933 and liquid index of refraction 1.333. The values considered at the end of the analyses were: mean diameter, standard deviation, PDI and Zeta potential.

TEM analysis was performed on samples of freshly prepared liposome suspensions. The samples were stained with uracil acetate to allow the analysis by TEM and images were acquired with magnifications of 50 K, 120 K, 150 K and 200 K by an electronic microscope JEOL JEM-1200EXIII provided with TEM CCD camera Mega View III.

The resulting images were processed with ImageJ software, a digital image processing computer program that can calculate liposomes’ diameter and area. The program is supported by standard image processing functions, such as logical and arithmetic operations between images, brightness and contrast adjustment, sharpening, smoothing and contour recognition. It also performs geometric transformations, such as scaling, rotation and reflection [32].

For the quantification of BSA-loaded liposomes, each batch (1 mL) was centrifuged to separate free BSA from encapsulated BSA. Centrifugation was performed at 16,400 rpm, 4 °C for 30 min. The obtained pellet was resuspended using 2% Triton-X in PBS at pH= 7.4 solution and left 5 min at 75 °C to achieve liposome disruption and BSA extraction. Samples of 25 µL were analyzed by UV spectrophotometer at 562 nm wavelength (Microplate Photometer MPP-96 BioSan^®^, BMG LABTECH, Freiburg). BSA quantification was performed by Pierce^®^ BCA protein assay kit (Thermo Fisher Scientific Inc., Rockford, IL, USA), following a method reported in the literature [33]. A calibration curve was drawn by UV spectrophotometric analysis at 562 nm wavelength, between 0.025 and 2 mg/mL BSA concentrations. BSA encapsulation efficiency was calculated according to Equation (1):(1)$$EE\%=\frac{C}{A}\times \;100$$
where *A* is the total amount of BSA or Myo added during preparation, *C* is the amount of BSA or Myo detected by UV spectrophotometric analysis and encapsulated in liposomes.

The same method was used to extract Myo loaded into liposomes, but, in this case, the solution containing the disrupted liposomes and myoglobin was directly analyzed by UV micro-plate reader at 410 nm wavelength. Myoglobin encapsulation efficiency was calculated applying Equation (1).

In vitro drug release tests for evaluating Myo release from liposomes obtained by both preparation methods were performed in PBS pH 7.4 at 37 °C. The preliminary in vitro release tests permitted selection of the most suitable liposome preparation method and, thus, the liposome batches to be further characterized. In vitro release tests in PBS at 4 °C, and in PERF-GEN™ solution at 4 °C, were performed on the selected liposome batches. All in vitro release tests were carried out as follows.

The pellets obtained from centrifugation (16,400 rpm, 30 min, 4 °C) of fresh liposome suspensions were resuspended in 1 mL PBS or PERF-GEN solution, and then added into a 100 kDa cut-off dialysis membrane tube that was soaked in 5 mL PBS or PERF-GEN solution and was kept under magnetic stirring at 4 °C and 37 °C for the fixed time. At defined times (30 min, 1 h, 2 h, 3 h, 4 h and 5 h), 1 mL PBS external buffer solution was withdrawn and replaced by an equal volume of fresh PBS (1 mL). Pure Myo solution was used as a control. The withdrawals were analyzed by UV-VIS spectrophotometer micro-plate reader (λ 410 nm). Each batch was analyzed in triplicate. Myoglobin release percentage was determined using the following formula in Equation (2):(2)$$Drug\;release\;\left(\%\right)=\;\frac{R_{t}}{L}\;\times \;100$$
where *L* represents the amount of drug loaded into liposomes and *R_t_* the cumulative amount of drug released at time *t*.

In vitro release tests for evaluating BSA release from liposomes obtained by both preparation methods were performed in PBS pH 7.4 at 37 °C. The preliminary in vitro release tests permitted selection of the most suitable liposome preparation method and , thus, the liposome batches to be further characterized. In vitro release tests in PBS at 4 °C, and in PERF-GEN™ solution at 4 °C, were performed on the selected liposome batches. All in vitro release tests were carried out as follows.

The pellets obtained from 1 mL centrifuged (30 min, 4 °C, 16,400 rpm) batch were resuspended in 1 mL of PBS buffer (pH 7.4) or 1 mL PERF-GEN solution into a safe-lock tube that was incubated at 37 °C or 4 °C in a shaker incubator. At defined times (1 h, 2 h, 3 h and 4 h), the samples were centrifuged (16,400 rpm, 30 min, 4 °C) and liposomes containing BSA precipitated. A total of 100 µL of PBS supernatant solution was withdrawn at each fixed time and replaced with the same amount of PBS or PERF-GEN solution. The samples were resuspended by 1 min vortex before replacing them in thermostatic conditions.

To quantify the BSA released, Pierce^®^ BCA protein assay kit was used. The 96-well plate was analyzed by UV-VIS spectrophotometer micro-plate reader (λ 562 nm) as reported above. Each batch was analyzed in triplicate. BSA released % was determined using Equation (2).

#### 4.2.3. Preparation of Fluorescent-Labeled Liposomes

Placebo liposomes manufactured with Nanoassemblr were formulated with a fluorescent phospholipid to study cellular uptake in an ex vivo model of harvested kidney. Fluorescein-DHPE (N-(Fluorescein-5-thiocarbamoyl)-1,2-dihexadecanoyl-sn-glycero-3-phosphoethanolamine, triethylammonium salt) was used as a fluorescent marker for liposomes due to its ability to be entrapped in the liposome’s double layer. Fluorescein-DHPE stock solution (1 mg/mL) was obtained by solubilizing 5 mg of fluorescent phospholipid in 5 mL of ethanol.

The preparation process was carried out as follows: DSPC and cholesterol were solubilized in 2.6 mL of ethanol and 0.39 mL of Fluorescein-DHPE stock solution was added, to obtain a lipids molar ratio (DSPC:Chol:Fluorescein-DHPE) of 49.5:49.5:1. Nanoassemblr software was set to mix the PBS buffer solution pH 7.4 and fluorescent lipidic ethanolic solution at FRR 3:1 and TFR 8 mL/min.

The fluorescent liposome batches were characterized for their particle size and polydispersity index (PDI) by DLS.

#### 4.2.4. Fluorescent-Labeled Liposomes In Vitro Test on Porcine Epithelial Tubular Renal Cells

Porcine tubular renal cells were used to study cellular uptake of fluorescent liposomes. The cells were seeded into 35 mm glass-bottom Petri dishes at a density of 600,000 cells in 2 mL of complete medium (DMEM, 10% FBS, 1% penicillin and streptomycin). Cells were left 24 h in incubation (37 °C, 5–6.5% CO_2_) to guarantee cell adhesion. Subsequently, the medium was replaced with fresh growth medium. A total of 300 µL of fluorescent fresh liposome suspension (0.25 mg/mL) was added to and analyzed for 4 h incubation by confocal laser scanning microscope (CLSM, Fluoview FV 10i (Olympus, Tokyo, Japan)). Particular points of the Petri dish were chosen and focused to acquire images of cells and liposomes every 10 min. Microscope image sequences were recorded and then viewed at a greater speed to give an accelerated view of the microscopic process (time-lapse). At the end of the analysis, z-stack images were acquired. Z-stack imaging is a compilation of photographs taken at a set interval between the first and last planes of focus of a pollen grain. These images are combined into a brief “Real-time” video that allows users to explore the pollen grain at any plane of focus without the use of a microscope.

#### 4.2.5. Fluorescent-Labeled Liposomes Ex Vivo Test on Harvested Pig Kidney

To simulate a circulatory death donation, the pig was subjected to warm ischemia of aorta for 30 min and the excised kidney was subsequently perfused for 4 h at 4 °C with PERF-GEN solution (IGL, Lissieu, France), supplemented with 2 mL of fluorescent-labeled liposomes (3 × 10^11^ particles/mL), using a machine perfusion (waves, IGL). A renal biopsy was performed at the end of the perfusion and the tissue was stored in 10% formalin for the histological analysis.

Confocal laser scanning microscopy (CLSM), Fluoview FV 10i (Olympus, Tokyo, Japan), was used to study fluorescent liposomes’ cellular uptake. CLSM is equipped with filter cube Y3ET (CY3 green). Kidney tissue slides obtained at t_0_ (before liposome perfusion) and at t_4_ (4 h after liposome perfusion) were analyzed.

## 5. Conclusions

The results reported and discussed in this work can be summarized for the different aspects involving liposome formulation/manufacturing and liposome behavior in non-physiological conditions. From the formulation and manufacturing process stand points, the results corroborate the advantages of the microfluidic technique for obtaining reproducible liposomes with suitable size below 200 nm. Moreover, it has been highlighted how protein encapsulation efficiency depends on their molecular weight and isoelectric point, and can be modulated accordingly.

Liposome behavior has been explored in the non-physiological conditions used in isolated kidney perfusion. To the best of the authors’ knowledge, protein release from liposomes in these conditions have never been considered yet. The results obtained show that, as expected, lowering incubation temperature slows down the proteins release. Protein release must be evaluated case by case in the perfusion fluid, as the results obtained demonstrated that release of proteins sensitive to ionic media (such as BSA) is affected by the perfusion medium.

Cellular uptake has been shown to happen in vitro. However, the poor cellular uptake highlighted in these preliminary results could be justified by the neutral surface charge of liposomes that delayed their interaction with cells which possess negative surface charge [28]. The future direction of the research is towards the use of ionizable lipids in liposome formulation.

The strength of our study is having tested the behavior of liposomes ex vivo under the real condition (hypothermia) of a perfused organ before transplantation. This is the first step to finding a suitable vehicle for the delivery of drugs to condition organs, increase the pool of donors and reduce the patients’ waiting time for a kidney transplantation.

## Figures and Tables

**Figure 1 ijms-23-07999-f001:**
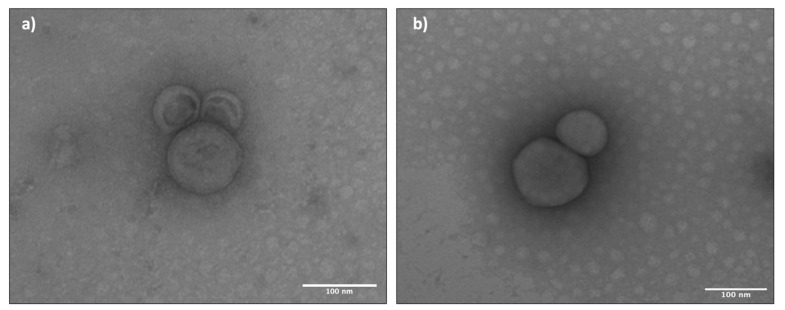
TEM images of liposomes: (**a**) batch #1 (120 K), (**b**) batch #2 (120 K).

**Figure 2 ijms-23-07999-f002:**
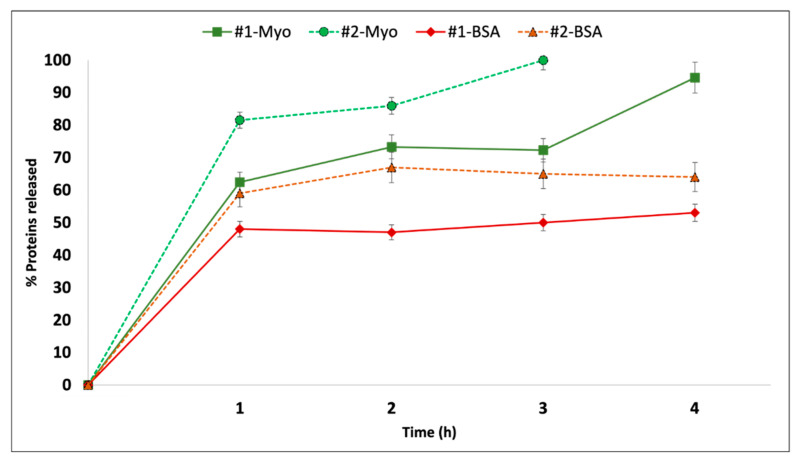
In vitro release of: BSA and Myo, from liposome batches obtained by THF (batches # 1 BSA, 1 Myo) and microfluidics (batches # 2 BSA, 2 Myo) at 37 °C in PBS.

**Figure 3 ijms-23-07999-f003:**
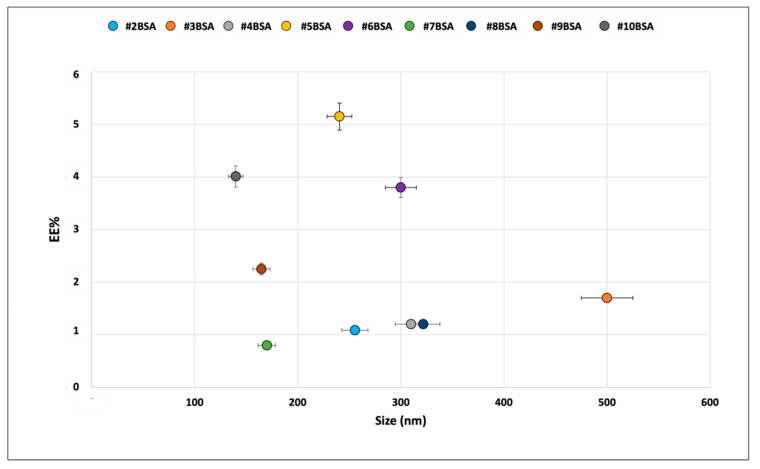
Average particle size determined by DLS and EE% of BSA-loaded liposomes obtained by modified formulation and process conditions.

**Figure 4 ijms-23-07999-f004:**
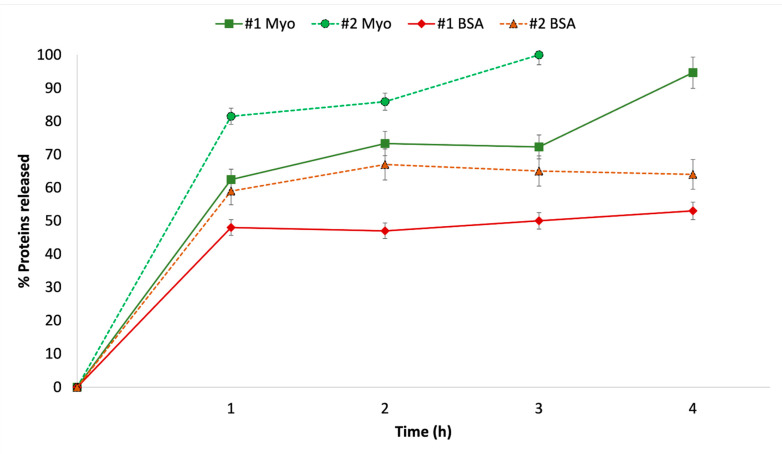
In vitro releases of BSA and Myo from liposomes manufactured with microfluidic technique, in different media (PBS, PERF-GEN) and temperature conditions (37 °C, 4 °C).

**Figure 5 ijms-23-07999-f005:**
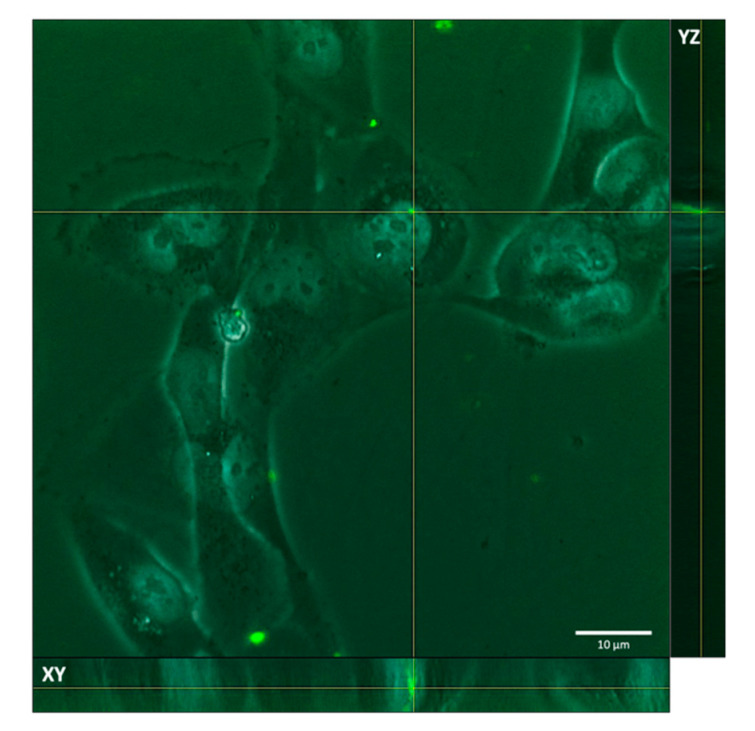
Z-stack of fluorescent liposomes co-incubated with tubular renal cells.

**Figure 6 ijms-23-07999-f006:**
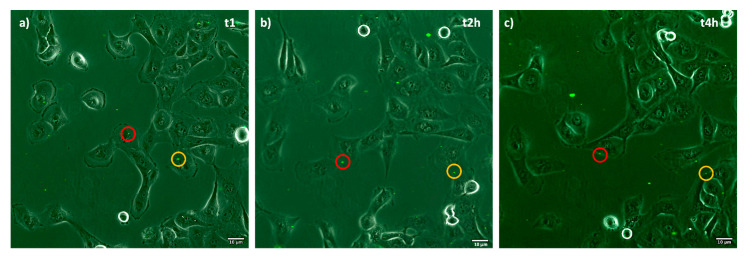
Time-lapse confocal laser scanning microscopy of fluorescent liposomes’ cellular uptake in tubular renal cells during 4 h incubation: (**a**) 1 h, (**b**) 2 h, (**c**) 4 h. The circle colors highlight the displacement of liposomes taken during the 4 h of incubation thus demonstrating their permanence in the cell body in this time progression: white, yellow and red.

**Figure 7 ijms-23-07999-f007:**
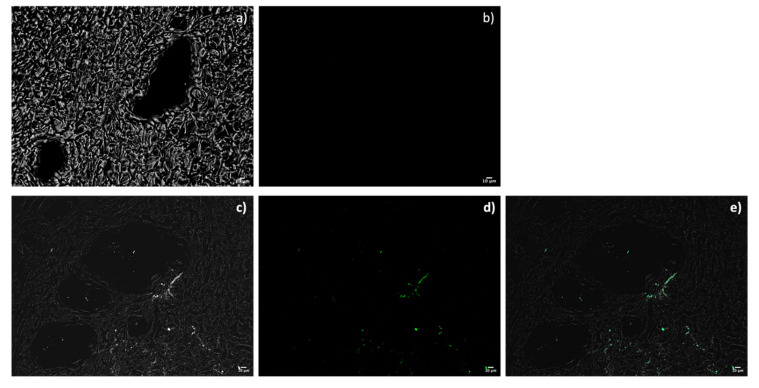
Fluorescent microscopy analysis of (**a**) kidney tissue t_0_; (**b**) kidney tissue t_0_ (Y3ET—CY3 green); (**c**) kidney tissue t_4_; (**d**) kidney tissue t_4_ (Y3ET—CY3 green); (**e**) merge kidney tissue t_4_.

**Figure 8 ijms-23-07999-f008:**
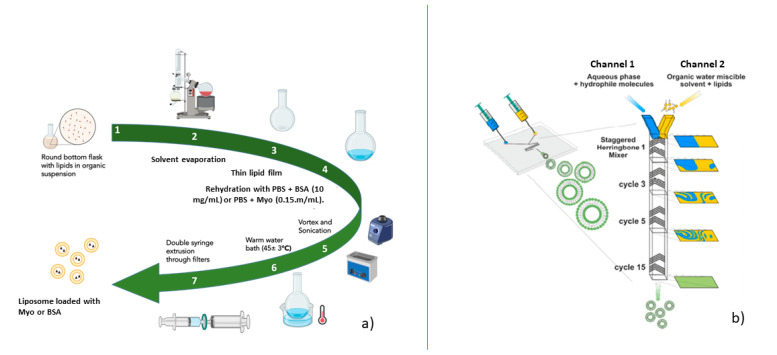
Schemes of protein-loaded liposome preparation methods: (**a**) thin film hydration, (**b**) microfluidic technique by Nanoassemblr device.

**Table 1 ijms-23-07999-t001:** Examples of liposome formulations in clinical trials.

Clinical Trial	Date	Conditions	Drugs	NCT Code
“Efficacy of liposomal bupivacaine for pain control after percutaneous nephrostolithotomy”	February 2017	-Renal calculi -Postoperative pain	-Liposomal Bupivacaine -Bupivacaine -Saline solution	NCT03043027
“Endovenous versus liposomal Iron in CKD”	May 2013	-Iron deficiency anemia -Chronic kidney disease	-Gluconate iron -Liposomal iron	NCT01864161
“A multicenter phase I study of MRX34, microrna mir-RX34 liposomal injection”	April 2013	-Primary liver cancer -SCLC -Lymphoma -Melanoma -Multiple myeloma -Renal cell carcinoma -NSCLC	-MRX34	NCT01829971
“TAP Blocks with ropivacaine continuous infusion catheters vs single dose liposomal bupivicaine after kidney transplant”	November 2018	-Transplant; kidney failure -Postoperative pain	-Ropivacaine continuous infusion catheter -Single-dose liposomal Bupivicaine	NCT03737604
“The LIPMAT study: liposomal prednisolone to improve hemodialysis fistula maturation”	July 2015	-Renal dialysis -Hemodynamics -Vascular remodeling -Neointima	-PEG-liposomal prednisolone sodium phosphate -Placebo	NCT02495662

**Table 2 ijms-23-07999-t002:** Average particle size, PDI, zeta potential and encapsulation efficiency of the liposome batches prepared with the two manufacturing techniques.

Batch #	Preparation Method	Molar Ratio DSPC:Chol	Average Particle Size (nm ± SD)	PDI	Zeta-Potential mV	Encapsulation Efficiency %
**1**	TFH	50:50	163.9 ± 3.2	0.23	+1.2 ± 0.5	NA
**1 BSA**	TFH	50:50	467.0 ± 32.5	0.76	+2.0 ± 0.2	0.9 ± 0.2
**1 Myo**	TFH	50:50	179.5 ± 5.1	0.39	-	67.4 ± 7.6
**2**	Microfluidic	50:50	112.9 ± 0.7	0.23	+1.1 ± 0.7	NA
**2 BSA**	Microfluidic	50:50	255.4 ± 17.9	0.54	−2.9 ± 0.8	1.10 ± 0.5
**2 Myo**	Microfluidic	50:50	139.8 ± 11.1	0.35	-	27.5 ± 8.5

**Table 3 ijms-23-07999-t003:** Results of TEM images processed with ImageJ software: diameter and circularity of batches #1 and 2, and comparison with liposome average diameter obtained by DLS.

Batches #	Diameter (nm) from ImageJ Elaboration of TEM	Diameter (nm) from DLS Analysis	Bilayer Thickness from TEM:DLS Diameter Ratio (nm)	Circularity
**1**	171.9 ± 15,3	163.9 ± 3.2	1.0 ± 0.1	0.84 ± 0.008
**2**	100.7 ± 12, 0	112.9 ± 0.7	0.9 ± 0.1	0.93 ± 0.004

**Table 4 ijms-23-07999-t004:** Modified formulation and process parameters applied to BSA loading into liposome manufacturing by microfluidic technique.

Batches #	DSPC:CHOL	TFR (mL/min)	Trehalose %	BSA Solution Composition
**3 BSA**	70:30	8	-	PBS
**4 BSA**	50:50	8	10	PBS
**5 BSA**	50:50	8	20	PBS
**6 BSA**	50:50	8	40	PBS
**7 BSA**	50:50	12	-	PBS
**8 BSA**	50:50	12	20	PBS
**9 BSA**	50:50	8	-	Purified water
**10 BSA**	50:50	8	20	Purified water

## Data Availability

Not applicable.

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
