# Peer review of "Liposome Formulation and In Vitro Testing in Non-Physiological Conditions Addressed to Ex Vivo Kidney Perfusion"

_ijms, 2022, doi:10.3390/ijms23147999_

Round 1
Reviewer 1 Report
This work has seemed very interesting to me, quite good results have been obtained and the different types of systems used have been well characterized. However, there are some errors that should not be overlooked as they are basic.
-There are many typographical errors, for example, in lines 69, 249, 257, 287, 293, 340, 354...
-Some drugs appear in lowercase (vancomycin) and others in uppercase (Prednisolone).
- In line 220 it's not clear if they are referring to an increase or decrease in temperature in reference to what makes Myo release go down.
- Review phrases such as in line 144 or 241.
- The scales in figure 1 are not clearly visible and almost have to be guessed.
- Errors and significant numbers must be checked and written correctly, both in the tables and in the text. Wrong examples: 163.9 ± 3.25; 467± 32.52 ; 179.5 ± 5.1 !!! .
Author Response
The point by point rsponse to reviewer 1 are uploaded here below.

Reviewer 2 Report
Scale bar of TEM images of liposomes on Fig 1. are very blurry, should be change the pictures or fit a clear scalebar.
Fig. 1 it would be good, to make a histogram of particles size and compare it to DLS. It is well known that the DLS is giving hydrodynamic radius. So, the difference can be proportional to the double layer thickness. Besides, some experience shown that the molecular weight characteristics of vesicular subunits result in that the bilayer thickness of independently assembled vesicles can be 3–7 nm for liposomes [1-3]. The layer thickness determined based on TEM photos is many times this. What could be the reason for this significant difference?
The quality of the figures showing the results of the in vitro release should be higher. The frame is not necessary, but the alignment of the axes is necessary. Curves and data points could be coloured.
In the case of Fig 3 the frame is not necessary, but the alignment of the axes is necessary.
Fig 5. is oversized.
References
[1] E. Rideau, R. Dimova, P. Schwille, F.R. Wurm, K. Landfester, Liposomes and polymersomes: a comparative review towards cell mimicking, Chem. Soc. Rev. 47 (2018) 8572–8610. https://doi.org/10.1039/c8cs00162f.
[2] K. Shimada, A. Miyagishima, Y. Sadzuka, Y. Nozawa, Y. Mochizuki, H. Ohshima, S. Hirota, Determination of the thickness of the fixed aqueous layer around polyethyleneglycol-coated liposomes, J. Drug Target. 3 (1995) 283–289. https://doi.org/10.3109/10611869509015957.
[3] Á. Juhász, D. Ungor, K. Berta, L. Seres, E. Csapó, Spreadsheet-based nonlinear analysis of in vitro release properties of a model drug from colloidal carriers, J. Mol. Liq. 328 (2021) 115405. https://doi.org/10.1016/j.molliq.2021.115405.
Author Response
The point by point response to reviewer 2 are uploaded here below.
